# Propagation Characteristics of Magnetic Tomography Method Detection Signals of Oil and Gas Pipelines Based on Boundary Conditions

**DOI:** 10.3390/s22166065

**Published:** 2022-08-13

**Authors:** Linlin Liu, Lijian Yang, Songwei Gao

**Affiliations:** 1School of Information Science and Engineering, Shenyang University of Technology, Shenyang 110870, China; 2School of Information and Control Engineering, Liaoning Petrochemical University, Fushun 113001, China

**Keywords:** magnetic tomography method, magnetic memory signal, boundary condition, propagation characteristics

## Abstract

The magnetic tomography method (MTM) is a non-contact external inspection method for detecting metal magnetic memory signals. It has great potential for application in long-distance oil pipeline and subsea pipeline inspection. However, the spatial distribution characteristics and propagation laws of magnetic signals are not yet clear, which makes the MTM passive detection. In this study, a three-dimensional mathematical model of the magnetic field distribution of the stress concentration zone outside the pipe was established based on the boundary conditions. For the two cases in which the stress concentration zone was located at the top and bottom of the inner wall of the pipe, the model was solved by finite element analysis. The variation law of the magnetic signal outside the pipe was analyzed, and experiments were designed to verify the model. The results show that the shape of the magnetic memory signal remained unchanged after passing through the pipe wall. As the magnetic permeability of the pipe medium is much larger than that of air, the magnetic memory signal is significantly attenuated after penetrating the pipe wall. As the detection height increases, the magnetic induction outside the tube decays exponentially. The results also prove that the magnetic tomography method can detect the stress concentration zone at any position of the pipeline, and the detection accuracy is higher when it is located at the top of the pipeline.

## 1. Introduction

Long-distance pipelines are subject to defects, corrosion, cracks, and other safety hazards during service. Oil and gas leaks can lead to significant losses to the national economy and serious environment pollution. Therefore, it is crucial to regularly test pipelines for safety. Nondestructive testing (NDT) methods that are based on the principle of metal magnetic memory [1,2] have been widely used in recent years for the safety inspection of oil and gas pipelines. However, because of the special structure and geographical layout of the pipeline, for example, where pipeline pigs are inaccessible [3,4], it is not easy to implement internal inspection; therefore, external inspection of the pipeline can be performed. Magnetic tomography method (MTM) detection [5], developed by the Russian Transkor-K Center, is an emerging method for non-contact external detection. Compared with the traditional internal detection method, the MTM is used to detect buried pipelines without digging the overlying soil. This method is currently used for the online inspection of buried and submarine pipelines [3,5,6].

The principle of MTM detection is based on the Villari effect, which refers to the fact that the magnetic domain structure inside a magnetic material is closely related to its stress state. The stress state of a magnetic material is closely related to its internal domain structure, and the change of the domain structure will directly lead to the change of the magnetic field properties of the material. A pipe is naturally magnetized by the geomagnetic field and the magnetic flux line is distorted at locations such as the stress concentration zone [7]. The magnetic field of a pipeline is recorded remotely by inspection equipment [8,9] as it moves along the pipeline [6]. The safety of the pipeline is assessed by analyzing the location and direction of the stress concentration zone in the inspection data; determining the type, location, and direction of defects; and classifying their danger. The principle of detection is shown in Figure 1. Li et al. predicted the distribution of geomagnetic leakage magnetic field (GMLF) based on the magnetic dipole theoretical model. The model did not consider the influence of the remaining part of the pipeline around the magnetized region on the distribution of the leakage field, resulting in high prediction values [10]. After modifying the model to include the pipe material, Jarvis et al. found that the GMLF was significantly lower than the original prediction [11]. To reduce the influence of the surrounding magnetic media on the MTM detection results, Li [12] evaluated the disturbance of the magnetic field around a pipeline that is caused by defects and ferromagnetic objects, designed an AMR sensor array to detect defects, and presented a relationship between the defect detection rate and distance. Liu studied the distribution of magnetic memory signals in a pipe based on the J-A model and analyzed the characteristics of magnetic memory signals in non-ferromagnetic and ferromagnetic substances. The model is based on saturation magnetization and is not applicable to the characterization of non-saturated magnetization signals, such as geomagnetic magnetization [13]. There is still a lack of relevant research on the propagation law and distribution characteristics of magnetic memory signals of the stress concentration zone outside a pipeline under weak magnetization. At present, most of the theoretical research on MTM detection is aimed at quantitatively analyzing the gradient modulus of the magnetic signal to identify the stress concentration zone, and the detection error is relatively high. To improve the accuracy of MTM detection, it is necessary to quantify the relationship between the stress concentration zone and detection signal. This study aims to address this issue.

In this paper, a spatial distribution model of the magnetic memory signal is established based on the magnetic dipole theory and the magnetic anisotropy Biot–Savart law [14], considering the medium distribution and boundary conditions in the process of magnetic signal transmission. Using the method of mutual verification between simulation calculation and experiment, the transmission law and distribution characteristics of the magnetic memory signal in different media are analyzed to explain the MTM detection mechanism.

## 2. MTM Modeling

The magnetic tomography method uses mathematical models to invert the spatial distribution of the stresses, mechanical loads, and structural changes in metals. The magnetic field outside a pipeline is a superposition of multiple physical fields. To simplify the analysis, the uncovered pipe was examined in this study without discussing the specific mathematical description of the defects. A magnetic dipole was used as a model for the magnetic field in the stress concentration zone. The distribution of the magnetic memory signal in the space outside the pipe of the stress concentration zone is modeled.

### 2.1. Magnetic Field Model of Stress Concentration Zone

It is assumed that the magnetic dipole is located on the xOy plane in the coordinate system Oxyz, where the y-axis is the axial direction of a pipe, the x-axis is the circumferential direction, and the z-axis is the radial direction, as shown in Figure 2.

The magnetic dipole is represented by a current-carrying coil [15,16]. There is a field point P in space; the vector radius is *r*, the radius of the current-carrying coil is a, and the vector radius from the source point to P is *R*. The angle between the current element *Id**l*** and the positive x-axis is φ. Equations (1) and (2) can be obtained as:(1)$$Idl=Idl\left(cos\varphi e_{y}-sin\varphi e_{x}\right)$$
(2)$$R=r-a=\left(rsin\theta -a\right)cos\varphi e_{x}+\left(rsin\theta -a\right)sin\varphi e_{y}+rcos\theta e_{z}$$

The distribution relation of free current is given by:(3)JdV=Idl
in which, J represents the free current density.

According to Maxwell’s equations, the constant magnetic field satisfies ∇·B=0, and ***B*** can be represented by the magnetic vector potential ***A*** as follows:(4)B=∇×A

It can be seen from the crystal magnetization curve [17] that the magnetization of ferromagnetic substances can be divided into easy and difficult magnetization directions. In the geomagnetic field, the magnetic field strength and magnetization direction in the most easily magnetized direction are the same as those in the external magnetic field, and the magnetization degree is the largest. To simplify the analysis, only the easy magnetization direction, that is, the main axis direction, was considered in this study. Then, the electromagnetic property in Equation (5) can be expressed as follows:(5)B=μH
where $\mu =\overset{3}{\underset{k=1}{\sum }}\overset{3}{\underset{i=1}{\sum }}\mu_{ki}{\vec{e}}_{k}{\vec{e}}_{i}$, $\mu_{ki}$ is the tensor permeability of the pipe.

When the main axis direction coincides with the three axes of the magnetic dipole coordinate system, only the main diagonal permeability of $\mu_{ki}$ is non-zero, that is, $\mu_{ki}$ ≠ 0, (*k* = *i* = 1, 2, 3). The integral form of the magnetic vector potential can be obtained as:(6)$$A_{\theta }^{\left(r\right)}=\frac{{∆}_{ki}}{4\pi \sqrt{\mu_{xx}\mu_{yy}\mu_{zz}}}\int^{\;}\frac{j_{\theta }\left(a\right)dV}{\sqrt{\frac{R_{x}^{2}}{\mu_{xx}}+\frac{R_{y}^{2}}{\mu_{yy}}+\frac{R_{z}^{2}}{\mu_{zz}}}}$$
where, $dV=da_{x}da_{y}da_{z}$, $R_{x}=\left(rsin\theta -a\right)cos\varphi$, $\;R_{y}=\left(rsin\theta -a\right)sin\varphi$, $R_{z}=rcos\theta$, ${∆}_{ki}=\left|\mu_{ki}\right|$·$\mu_{xx}$, $\mu_{yy}$, and $\mu_{zz}$ are the magnetic permeability components in the x, y, z directions, respectively.

The excitation magnetic source is a constant current-carrying coil; therefore, the magnetic field strength ***H*** satisfies Equation (7):∇ × ***H*** = ***J***(7)

From Equations (5) and (7) we obtain (8):(8)$$\nabla \times H=\nabla \times \mu^{-1}B=J\;$$

Substituting Equation (4) into Equation (8), Poisson’s Equation (9) for the magnetic vector potential ***A*** is obtained:(9)$$\nabla \left(\nabla \cdot A\right)\nabla^{2}A=-\mu J$$ (9) satisfies the Lorentz criterion and can be written as:(10)$$\nabla^{2}A-\frac{1}{c^{2}}\frac{\partial^{2}A}{\partial t^{2}}=-\mu J$$

### 2.2. Spatial Distribution Model

If the interference field is not considered, the spatial field containing the pipe under the action of a magnetic source can be divided into three parts. They are the inner cavity S_1_, pipe body S_2_, and spatial domain S_3_ from the outer wall of the pipe to infinity, as shown in Figure 3, where P is any point in S_3_ and the magnetic source is the stress concentration zone.

The propagation path of the magnetic memory signal is such that the magnetic signal in the stress concentration zone in S_1_ passes through S_2_, that is, the pipe body, and then passes through the outer wall of the pipe to S_3_. The propagation process passes through the ferromagnetic and then into air. The distribution of magnetic field lines and the intensity of the magnetic field change owing to changes in the medium [18]. Therefore, a boundary problem must be considered in this model.

The current-carrying coil was assumed to be placed at the top of the inner wall of the pipe. The boundary between the inner cavity and the inner wall of the pipe was set as the first interface, and the boundary between the outer wall of the pipe and outer space was the second interface. In other words, the first interface is between S_1_ and S_2_, and the second interface is between S_2_ and S_3_. According to the connection conditions of the magnetic field at the interface of different media, the following relationships were obtained:(11)$$n\times \left(H_{1}-H_{0}\right)=K$$
(12)$$n\times \left(H_{2}-H_{1}\right)=0$$
***n*** is the unit normal component of medium 1 to medium 2 and K is the conduction current density at the interface.

Substituting Equations (1) and (2) into Equations (11) and (12), respectively, Equations (13) and (14) are obtained:(13)$$n\times \left(\frac{1}{\mu_{\theta \theta }}\nabla \times A_{1}-\frac{1}{\mu_{0}}\nabla \times A_{0}\right)=K$$
(14)$$n\times \left(\frac{1}{\mu_{0}}\nabla \times A_{2}-\frac{1}{\mu_{\theta^{\prime }\theta^{\prime }}}\nabla \times A_{1}\right)=0$$
in which, $\mu_{\theta \theta }$ is the magnetic permeability of boundary 1, and $\mu_{\theta^{\prime }\theta^{\prime }}$ is the magnetic permeability of boundary 2.

The Coulomb norm (15) is satisfied between fields S_1_ and S_2_:(15)$$\nabla^{2}A_{1n}=\nabla^{2}A_{2n}=0$$

According to Equation (4), it can be deduced that the normal and tangential components of the magnetic flux intensity in the stress concentration zone [19,20] on the inner wall of the pipeline are as follows:(16)$$B_{in_{1n}}=\frac{aI}{4\pi }\sqrt{\frac{\mu_{1zz}}{\mu_{1xx}\mu_{1yy}}}\int^{\;}\frac{a-rsin\theta cos\varphi }{{\left\{\frac{{\left[\left(rsin\theta -a\right)cos\varphi \right]}^{2}}{\mu_{xx}}+\frac{{\left[\left(rsin\theta -a\right)sin\varphi \right]}^{2}}{\mu_{yy}}+\frac{{\left(rcos\varphi \right)}^{2}}{\mu_{zz}}\right\}}^{3/2}}d\varphi \;\;$$
and
(17)$$B_{in_{1t}}=\frac{aI}{4\pi }\sqrt{\frac{\mu_{1xx}}{\mu_{1yy}\mu_{1zz}}}\int^{\;}\frac{rcos\theta cos\varphi }{{\left\{\frac{{\left[\left(rsin\theta -a\right)cos\varphi \right]}^{2}}{\mu_{xx}}+\frac{{\left[\left(rsin\theta -a\right)sin\varphi \right]}^{2}}{\mu_{yy}}+\frac{{\left(rcos\varphi \right)}^{2}}{\mu_{zz}}\right\}}^{3/2}}d\varphi \;$$

Equation (5) can be expressed as Equation (18):(18)$$B_{s}=\sum_{i=1}^{3}\mu_{si}H_{i\;}\left(s=1,\;2,\;3\right)$$

Therefore, the normal and tangential component of the stress concentration zone on the outer wall of the pipeline can be expressed as:(19)$$B_{out_{1n}}=B_{1n}=\frac{aI}{4\pi }\sqrt{\frac{\mu_{1zz}}{\mu_{1xx}\mu_{1yy}}}\int^{\;}\frac{a-rsin\theta cos\varphi }{{\left\{\frac{{\left[\left(rsin\theta -a\right)cos\varphi \right]}^{2}}{\mu_{xx}}+\frac{{\left[\left(rsin\theta -a\right)sin\varphi \right]}^{2}}{\mu_{yy}}+\frac{{\left(rcos\varphi \right)}^{2}}{\mu_{zz}}\right\}}^{3/2}}d\varphi \;$$
(20)$$B_{out_{1t}}=\frac{aI}{4\pi }\mu_{1}\mu_{2}^{-1}\sqrt{\frac{\mu_{1xx}}{\mu_{1yy}\mu_{1zz}}}\int^{\;}\frac{a-rsin\theta cos\varphi }{{\left\{\frac{{\left[\left(rsin\theta -a\right)cos\varphi \right]}^{2}}{\mu_{xx}}+\frac{{\left[\left(rsin\theta -a\right)sin\varphi \right]}^{2}}{\mu_{yy}}+\frac{{\left(rcos\varphi \right)}^{2}}{\mu_{zz}}\right\}}^{3/2}}d\varphi \;$$
where, $\mu_{2}^{-1}=\frac{1}{\mu_{2xx}}e_{x}e_{x}+\frac{1}{\mu_{2yy}}e_{y}e_{y}+\frac{1}{\mu_{2zz}}e_{z}e_{z}$.

If the current-carrying coil [21] is placed on the bottom of the inner wall of the pipe, the propagation path of the magnetic memory signal in the stress concentration zone is first through the air in the inner cavity of the pipe, then through the pipe, and then out of the pipe. The equivalent magnetic field model in the stress concentration zone in air can be expressed as follows:(21)$$B_{in}=\frac{aI\mu_{0}}{4\pi }\int_{0}^{2\pi }\frac{rcos\theta cos\varphi e_{x}+rcos\theta sin\varphi e_{y}+\left(a-rsin\theta cos\varphi \right)e_{z}}{R^{3}}d\varphi$$

According to the superposition theorem of the magnetic field, it can be deduced that the normal and tangential magnetic flux intensity component on the top of the inner wall of the pipe are expressed as (22) and (23), respectively:(22)$$B_{in_{2n}}=\frac{aI\mu_{0}}{4\pi }\int_{0}^{2\pi }\frac{\left(a-rsin\theta cos\varphi \right)}{R^{3}}d\varphi +\frac{aI}{4\pi }\sqrt{\frac{\mu_{1zz}}{\mu_{1xx}\mu_{1yy}}}\int^{\;}\frac{a-rsin\theta }{{\left\{\frac{{\left[\left(rsin\theta -a\right)cos\varphi \right]}^{2}}{\mu_{xx}}+\frac{{\left[\left(rsin\theta -a\right)sin\varphi \right]}^{2}\;}{\mu_{yy}}+\frac{{\left(rcos\varphi \right)}^{2}}{\mu_{zz}}\right\}}^{3/2}}d\varphi$$
(23)$$B_{in_{2t}}=\frac{aI\mu_{0}}{4\pi }\int_{0}^{2\pi }\frac{rcos\theta cos\varphi }{R^{3}}d\varphi +\frac{aI}{4\pi }\sqrt{\frac{\mu_{1xx}}{\mu_{1yy}\mu_{1zz}}}\int^{\;}\frac{rcos\theta cos\varphi }{{\left\{\frac{{\left[\left(rsin\theta -a\right)cos\varphi \right]}^{2}}{\mu_{xx}}+\frac{{\left[\left(rsin\theta -a\right)sin\varphi \right]}^{2}}{\mu_{yy}}+\frac{{\left(rcos\varphi \right)}^{2}}{\mu_{zz}}\right\}}^{3/2}}d\varphi$$

The normal and tangential magnetic flux intensity components at any point on the outer wall of the pipe are as follows:(24)$$B_{out_{2n}}=\frac{aI\mu_{0}}{4\pi }\int_{0}^{2\pi }\frac{\left(a-rsin\theta cos\varphi \right)}{R^{3}}d\varphi +\frac{aI}{4\pi }\sqrt{\frac{\mu_{1zz}}{\mu_{1xx}\mu_{1yy}}}\int^{\;}\frac{a-rsin\theta }{{\left\{\frac{{\left[\left(rsin\theta -a\right)cos\varphi \right]}^{2}}{\mu_{xx}}+\frac{{\left[\left(rsin\theta -a\right)sin\varphi \right]}^{2}}{\mu_{yy}}+\frac{{\left(rcos\varphi \right)}^{2}}{\mu_{zz}}\right\}}^{3/2}}d\varphi$$
$$B_{out_{2t}}=\frac{aI}{4\pi }{\vec{\vec{\mu }}}_{1}{\vec{\vec{\mu }}}_{2}^{-1}$$
(25)$$\left\{\frac{aI\mu_{0}}{4\pi }\int_{0}^{2\pi }\frac{rcos\theta cos\varphi }{R^{3}}d\varphi +\frac{aI}{4\pi }\sqrt{\frac{\mu_{1xx}}{\mu_{1yy}\mu_{1zz}}}\int^{\;}\frac{rcos\theta cos\varphi }{{\left\{\frac{{\left[\left(rsin\theta -a\right)cos\varphi \right]}^{2}}{\mu_{xx}}+\frac{{\left[\left(rsin\theta -a\right)sin\varphi \right]}^{2}}{\mu_{yy}}+\frac{{\left(rcos\varphi \right)}^{2}}{\mu_{zz}}\right\}}^{3/2}}d\varphi \right\}$$

## 3. Simulation Calculation of Spatial Propagation Model

A finite element analysis method [22] was used to verify the spatial distribution model that was proposed in Section 2. The model consisted of a current-carrying coil and a pipe segment. The current-carrying coil was arranged on the inner wall of the pipe with a current density I of 10 A, and a diameter of 10 cm. The length, inner diameter, and wall thickness of the pipe were 6 m, 35 cm, and 2 cm, respectively. The values of the parameters that were used in the FEM model are shown in Table 1. The B-H curve of the pipe model is shown in Figure 4. The outside of the pipe was set to air, the air permeability is *μ*_0_ = 4π × 10^−7^ N/A^2^, and the pipe magnetic permeability was generated during the magnetization simulation.

The boundary conditions of the field are different; the magnetic memory signal propagation characteristics are also different because the stress concentration zone is located at different parts of the pipe. Therefore, the current-carrying coil was set at the bottom and top of the inner wall of the pipe, as shown in Figure 5a,b, respectively, to analyze the propagation law of the magnetic signal in the stress concentration zone at different locations and to explain the detection capability and detection range of MTM for stress concentration zones at different locations.

### 3.1. Propagation Characteristics of MTM Signal When the Stress Concentration Zone Is Located on the Top of the Inner Wall of the Pipe

The coordinate origin O in Figure 5a is taken as the center, where point O coincides with the center of the coil and scans along the y-axis direction. To show the change of the magnetic signal propagation more intuitively, the image display data are a solution of y = −100–100 mm. The magnetic flux intensity components under different lift-off values of H are shown in Figure 6.

As shown in Figure 6, the magnetic flux intensity on the pipe wall was the largest, and as the propagation distance of the magnetic memory signal outside the pipe increased, the magnetic flux intensity decreased. In Figure 6a, the tangential component crosses the zero point and the distance between the crest and trough is approximately 12 mm, which corresponds to the model width of the stress concentration zone. The magnetic memory signals that were obtained at different lift-off values maintained similar distribution trends and shapes. In Figure 6b, the signal peaks of the normal component of the magnetic flux intensity at different propagation distances are all at origin O, which coincides with the center of the stress concentration zone model and has the same shape.

Table 2 lists the peak values of the tangential and normal components of the magnetic flux intensity. To better analyze the attenuation and distribution characteristics of the signal, the peak differences of the normal components of the magnetic flux intensity that were detected at different lift-off values were compared, and the lift-off values differences were all 5 cm, as shown in Figure 7.

It can be seen from the peak difference that the magnetic flux intensity decreases the most between the pipe wall and 5 cm, indicating that the magnetic memory signal undergoes significant attenuation during the propagation process from the ferromagnetic medium to the air medium, which is also due to the difference that is caused by the different permeability of the medium. In the three ranges of 5–10 cm, 10–15 cm, and 15–20 cm, the decay of the magnetic signal was relatively flat, with Δ values less than 0.7. The decay trend of the peak normal component was analyzed via numerical fitting, and it was observed that the magnetic flux intensity decayed in a negative exponential pattern as the propagation distance increased.

### 3.2. Propagation Characteristics of MTM Signal When the Stress Concentration Zone Is Located at the Bottom of the Inner Wall of the Pipe

When the current-carrying coil is located at the bottom of the inner wall of the pipe, the center of the coil is O’, the x’-axis is parallel to the x-axis, and the y’-axis is parallel to the y-axis, as shown in Figure 5b. The scanning path of the detection sensor was the same as that set in Part 1 of Section 3. The distribution of the magnetic memory signal outside the pipe is shown in Figure 8.

As shown in Figure 8, when the coil is located at the bottom of the inner wall of the pipe, the magnetic flux intensity is attenuated by 10^3^ orders of magnitude compared with when the coil is on the top. The magnetic signal first passes through the air in the pipe cavity, and the magnetic permeability of the air is much smaller than that of the pipe; therefore, the signal undergoes large attenuation. Simultaneously, the signal is no longer a smooth curve as shown in Figure 6, but has many burrs, which also means that the signal is weaker and more susceptible to interference. These features require equipment with a high detection accuracy. In Figure 8a, as the propagation distance increases and the tangential component attenuates; however, the difference is not as obvious as that in Figure 5a. The tangential component has a peak and trough, and the location of the zero point is consistent with the center of the stress concentration zone, which has the same morphology and distribution trend as Figure 5a. As shown in Figure 8b, the normal component still exhibits a large attenuation in the propagation range from the pipe wall to 5 cm. Peaks at different propagation distances appear in the center of the stress concentration zone, and their shapes and trends are generally consistent.

To analyze the law by which the magnetic flux intensity decays with the propagation distance, the peak values of magnetic flux intensity under different lift-offs are given in Table 3.

From Figure 9, it can be concluded that for the case where the stress concentration zone is located at the bottom of the inner wall, the normal component decays with increasing propagation distance, and the change in the peak difference with a propagation interval of 5 cm is relatively flat. By fitting the peaks with the Levenberg–Marquardt algorithm, it can be seen that the peak decay of the flux intensity follows an exponential law with an increase in the lift-off value.

## 4. Experiment

A detection experimental platform was designed to verify the validity of the model. To prevent the interference of magnetic substances with the test results in the experimental environment, the pipe segment was placed in a relatively open field. The experiment considered only the influence of the pipe material and air medium on the propagation characteristics of the magnetic memory signal. The geomagnetic field value of the site was measured before the experiment and the detected data were processed after the experiment. The experimental platform consisted of a regulated power supply, current-carrying coil, pipe segment, detection sensor [23], host computer, and NI data processing system.

### 4.1. Design of Experiment

The experimental object was a conventional pipe, X80, with a length of 4 m, an inner diameter of 70 cm, and a wall thickness of 2 cm. A self-made DC magnetized coil with a diameter of 10 cm and a current density of 10 A was used. The detection sensor that was used in the experiment was a fluxgate sensor. The output is digital with a range of ±2mT, accuracy of 1 nT, and a sensitivity of 0.5 nT. The lift-off value H was set as 0, 5, 10, 15, and 20 cm, respectively. The scanning path was 2 m along the axial direction of the pipe, that is y = −1–1 m. The current-carrying coil was placed at the top (position 1) and bottom (position 2) of the inner wall of the pipe, respectively, and the center was located at the origin O of the coordinate system. The principle of the detection experiment is shown in Figure 10.

### 4.2. Experimental Results and Analysis

First, the coil was placed at position 1 in Figure 10, and the distribution of the tangential and normal components of the magnetic flux intensity that were measured at different lift-offs above the pipe is shown in Figure 11. It can be seen that the distribution characteristics of the measured signals are basically consistent with the theoretical analysis results.

The position where the peak value of the normal component that was measured in the experiment appears, as shown in Figure 12, is the difference between the peak distance that was measured in the experiment and the peak position of the theoretical data (horizontal axis 100 mm) under different lift-offs can be intuitively seen. The maximum difference was 8%, indicating that the distribution model of the magnetic flux intensity at the top was reliable.

Second, the experiment was carried out according to position 2 in Figure 10, and the measured magnetic flux intensity is shown in Figure 13 and Figure 14. Compared with Figure 8, the magnetic flux intensity component has the same distribution trend as the model result. By fitting the peak value of the normal component, it can be seen that the peak value still exhibits an exponential decay law as the propagation distance increases, indicating that the measurement results are consistent with the model calculation results.

The ratio of the actual detected value deviating from the theoretical origin of the stress concentration zone is 13% on the pipe outer wall, 7% for a 5 cm lift-off, and 39% for a 15 cm lift-off. For the magnetic field distribution of the stress concentration zone at the bottom of the inner wall of the pipe, the measured value deviates significantly from the model solution result. The signal is significantly attenuated due to the air layer in the pipe cavity, which affects the accuracy of the measurement. It can be observed that the stress concentration zone is located at different positions of the pipeline, and the accuracy of the measurement results of the magnetic field is different. It can be concluded from the comparison between theory and experiment that the calculated results of the model that is proposed in this paper is consistent with the characteristic distribution of the experimental measurements, indicating that the model can effectively describe the distribution characteristics of the magnetic signal of the stress concentration zone outside the pipeline.

## 5. Conclusions

To analyze the transmission law and distribution characteristics of the magnetic memory signal outside the pipeline and to qualitatively predict the position, shape, and propagation law of the stress concentration zone, a spatial distribution model of magnetic field of the stress concentration zone is modeled. Through a comparative analysis of the solution results of the model and experimental measurements, the following conclusions were drawn.

During the transmission of the magnetic memory signal from the pipe to outer space, the signal strength decays exponentially with an increase in the propagation distance. As the magnetic permeability of the pipe medium is much higher than that of air, the magnetic signal intensity decays at the highest rate in a small area from the pipe outer wall to the outside of the pipe.The characteristics of the stress concentration zone remain unchanged during the transmission of the magnetic memory signal. In other words, the normal component has a peak value and tangential component zero-crossing, and the peak and zero positions both appear in the central axis of the stress concentration zone. Therefore, the position and shape of the stress concentration zone can be determined from the detection signal. The detection mechanism of magnetic tomography was also explained.The magnetic memory signal is measurable outside the pipe, regardless of whether the stress concentration zone is located at the top or bottom of the pipe. It has been proved that the magnetic tomography method can measure the stress concentration zone at any position in a pipeline. By studying the magnetic field distribution of the current-carrying coils that are placed at different positions, the results show that the detection accuracy is higher when the stress concentration zone is located on the top of the inner wall of the pipeline. Therefore, the corresponding relationship between the position of the stress concentration zone in the pipeline and the detection accuracy is also a future research direction.

## Figures and Tables

**Figure 1 sensors-22-06065-f001:**
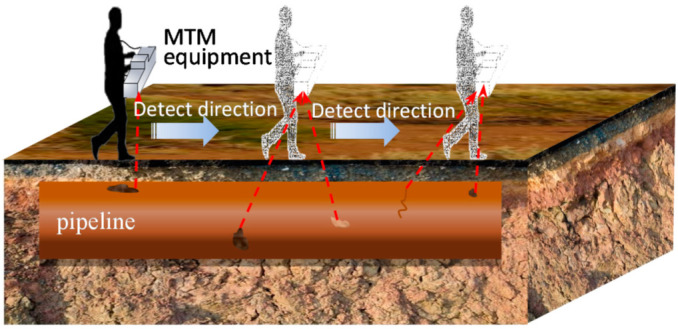
Schematic diagram of magnetic tomography method detection.

**Figure 2 sensors-22-06065-f002:**
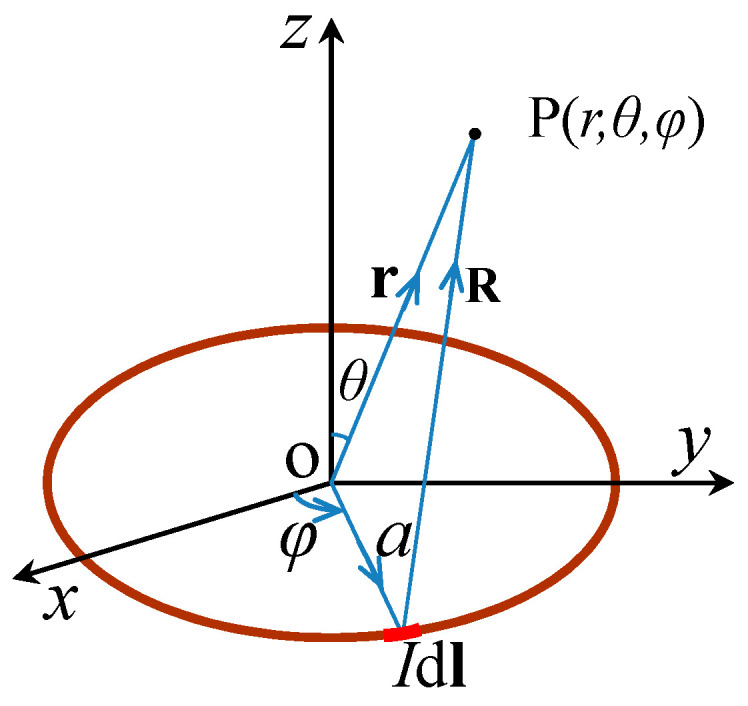
Stress concentration zone model.

**Figure 3 sensors-22-06065-f003:**
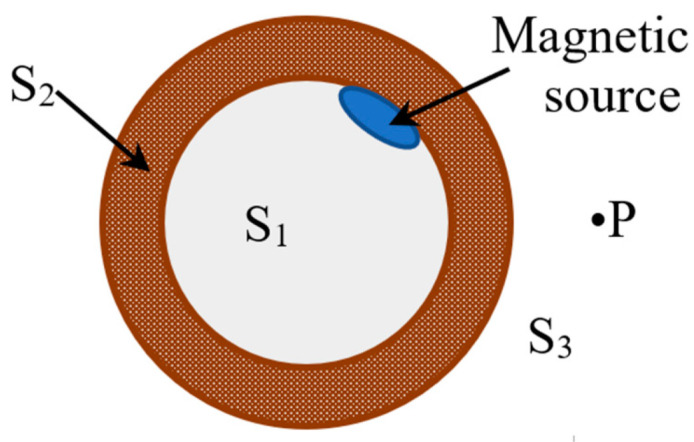
Field decomposition.

**Figure 4 sensors-22-06065-f004:**
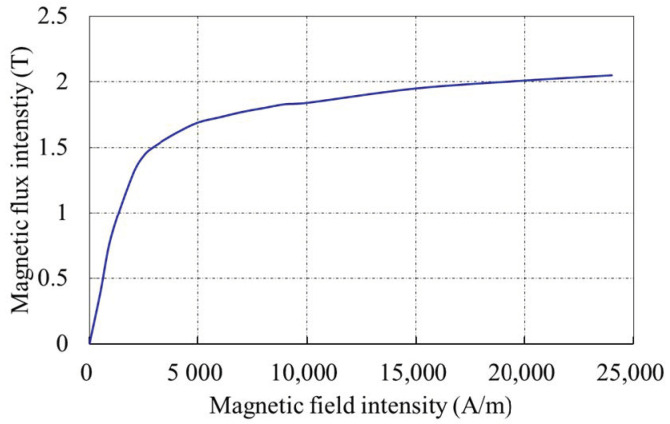
B-H curve of the pipe model.

**Figure 5 sensors-22-06065-f005:**
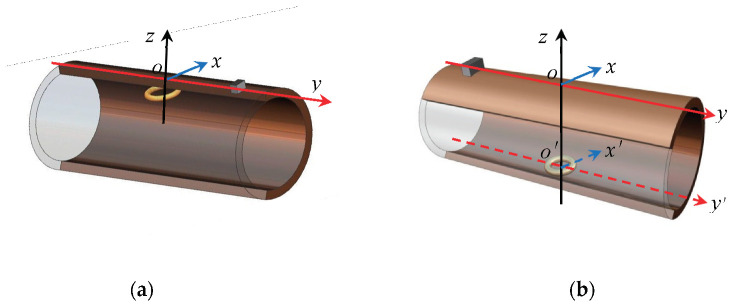
Position of the current-carrying coil. (**a**) On the top of the inner wall of the pipe. (**b**) At the bottom of the inner wall of the pipe.

**Figure 6 sensors-22-06065-f006:**
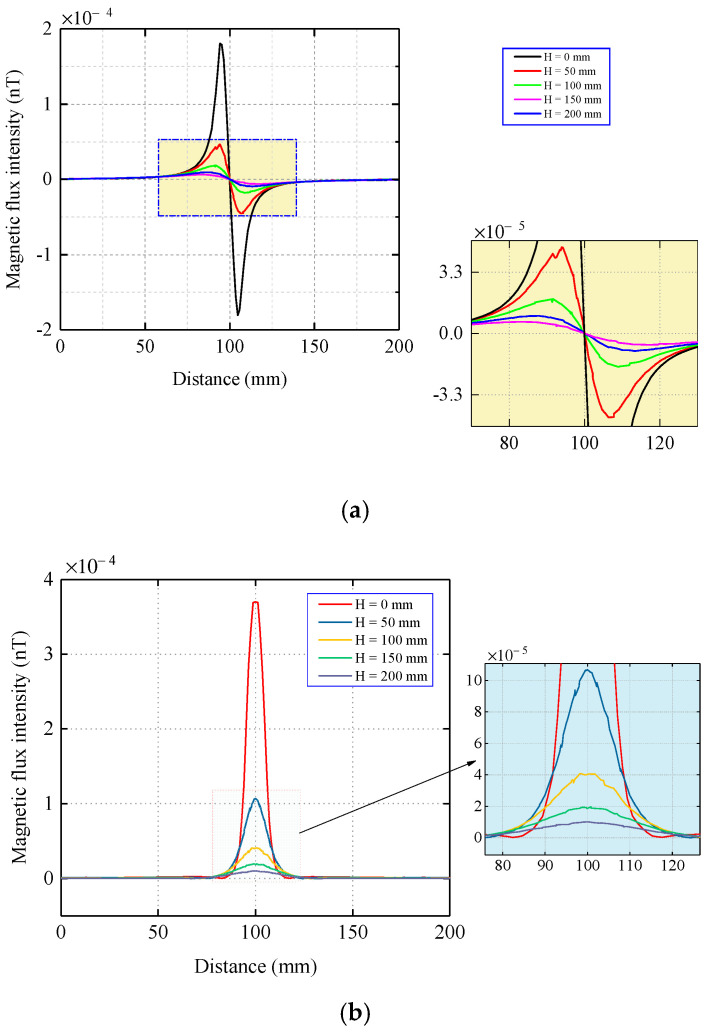
Distribution of the magnetic flux intensity when the stress concentration zone is located on the top of the inner wall of the pipe. (**a**) Tangential component. (**b**) Normal component.

**Figure 7 sensors-22-06065-f007:**
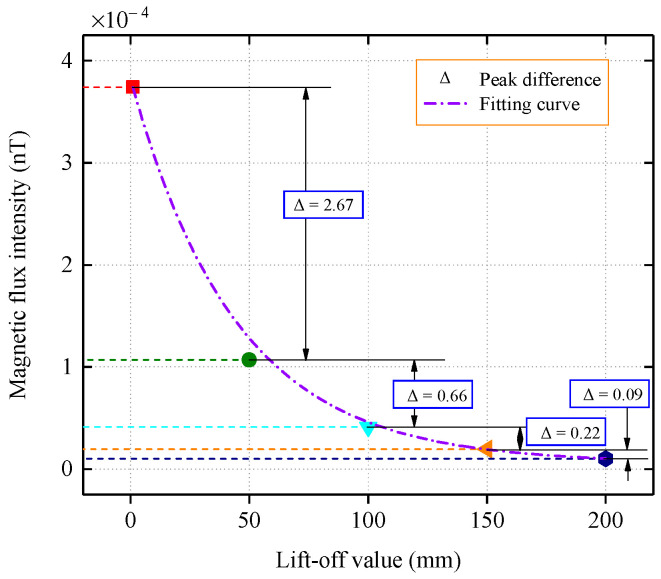
Normal component peak contrast.

**Figure 8 sensors-22-06065-f008:**
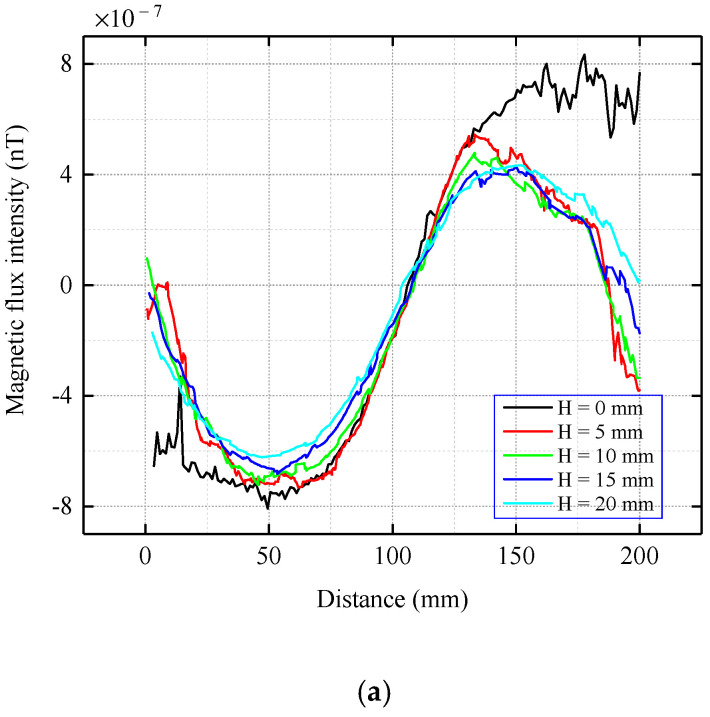
Distribution of the magnetic flux intensity when the stress concentration zone is located at the bottom of the inner wall of the pipe. (**a**) Tangential component. (**b**) Normal component.

**Figure 9 sensors-22-06065-f009:**
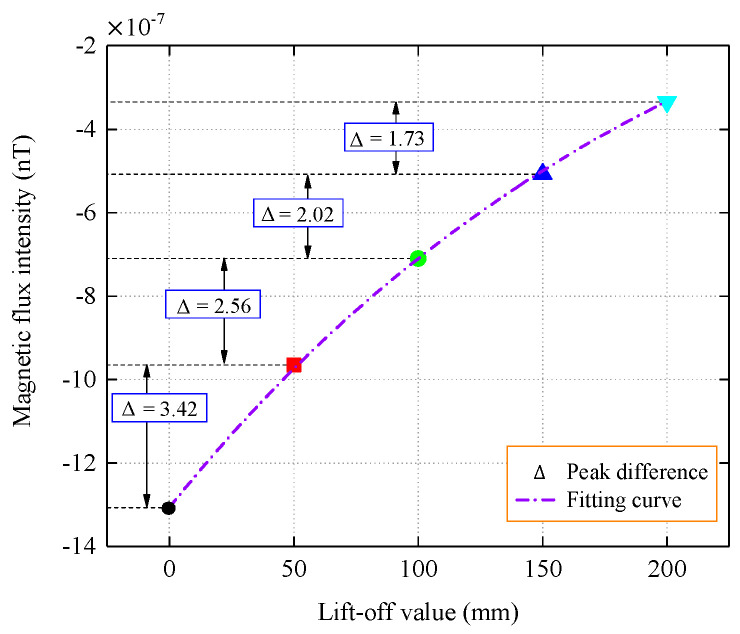
Peak value and trend fitting of normal component.

**Figure 10 sensors-22-06065-f010:**
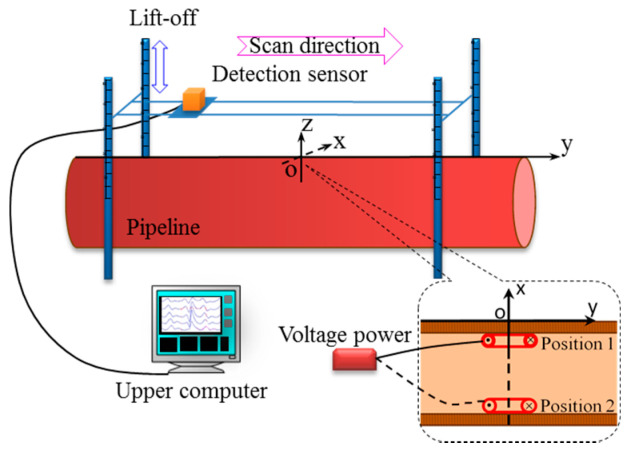
Experimental principle.

**Figure 11 sensors-22-06065-f011:**
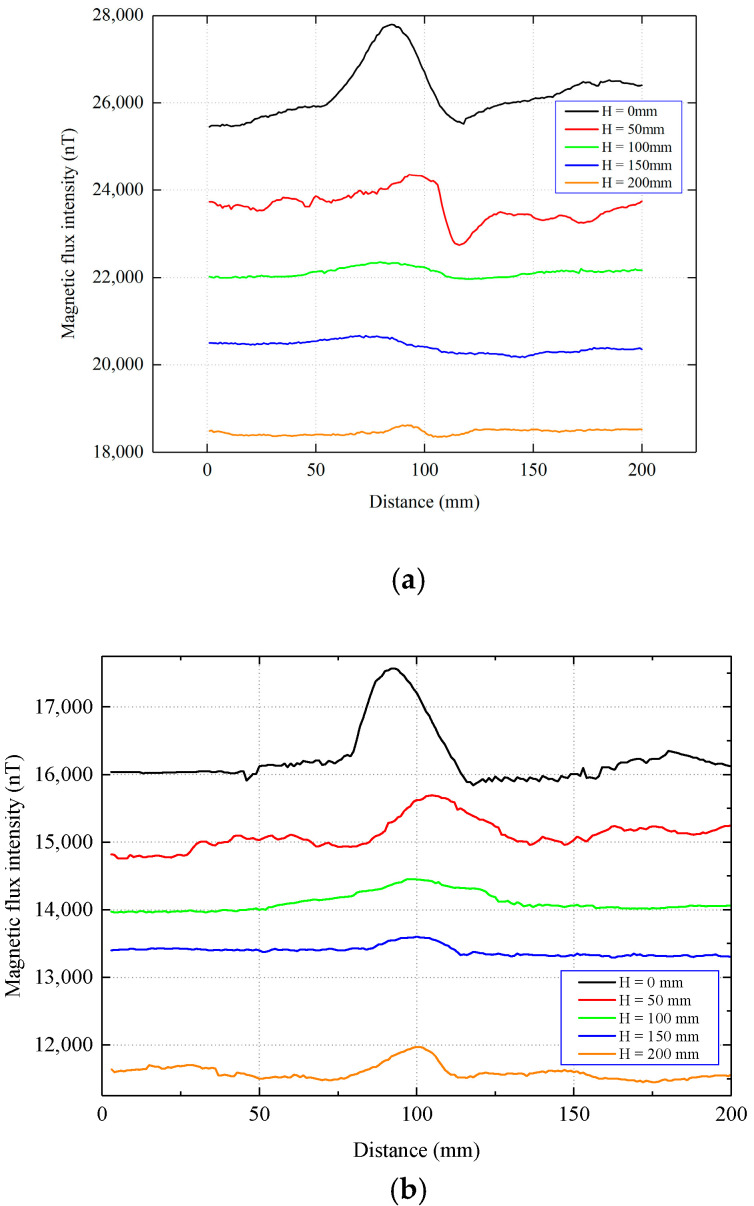
Experimental measurement of the magnetic flux intensity at different lift-offs. (**a**) Tangential component. (**b**) Normal component.

**Figure 12 sensors-22-06065-f012:**
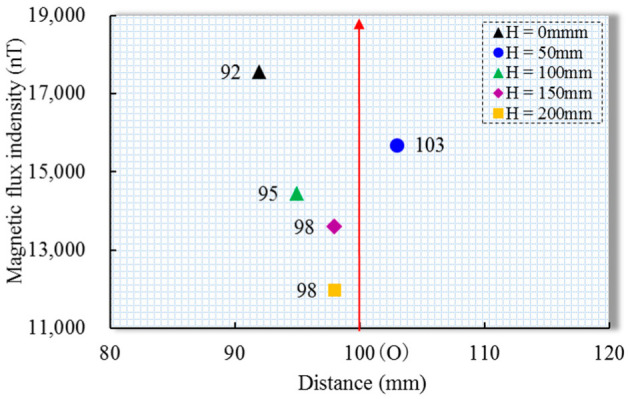
Comparison of the location of the peak of the measured normal component and the center of the stress concentration zone at different lift-offs.

**Figure 13 sensors-22-06065-f013:**
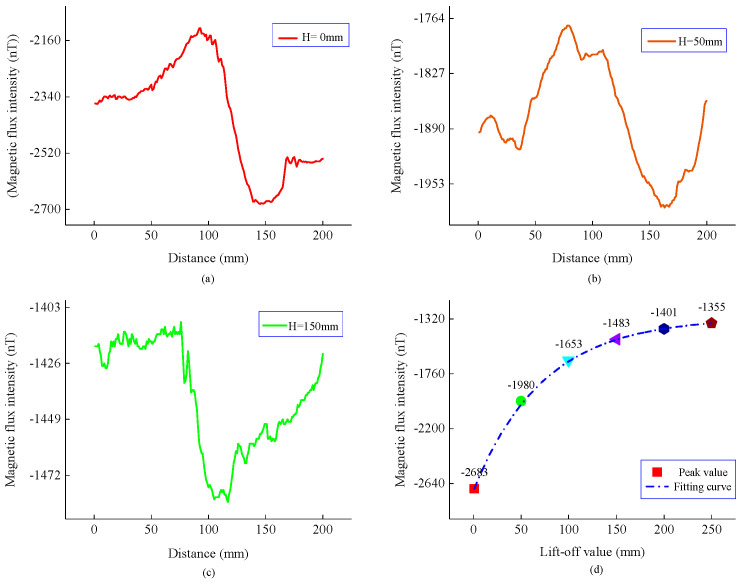
Normal components and the peak curve of the magnetic flux intensity at different lift-offs. (**a**) Normal components when H is 0 mm. (**b**) Normal components when H is 50 mm. (**c**) Normal components when H is 150 mm. (**d**) Peak and fitting curve.

**Figure 14 sensors-22-06065-f014:**
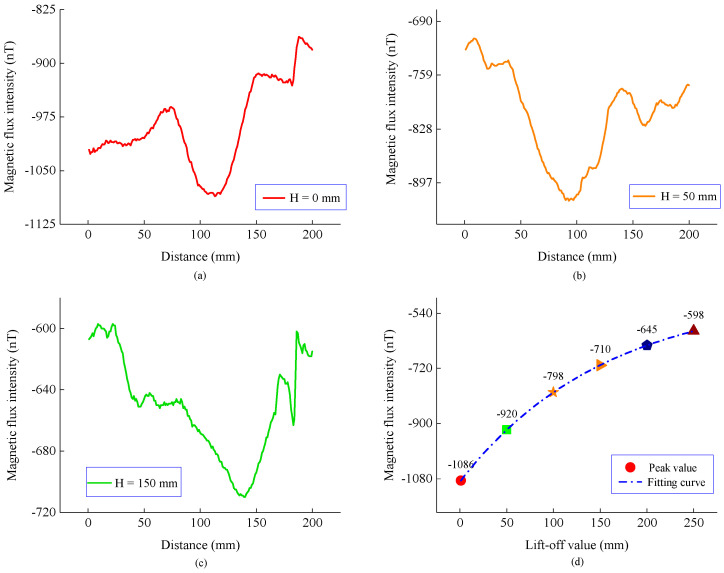
Tangential components and the peak curve of the magnetic flux intensity at different lift-offs. (**a**) Normal components when H is 0 mm. (**b**) Normal components when H is 50 mm. (**c**) Normal components when H is 150 mm. (**d**) Peak and fitting curve.

**Table 1 sensors-22-06065-t001:** Values of the parameters that were used in the FEM model.

Parameters	Value	Unit
Pipe diameter	700	mm
Wall thickness	20	mm
Length of pipe	6	m
Electrical conductivity	$4\times 10^{6}$	S/m
Pipe material	soft iron	-
Coil current density	10	A
Coil diameter	100	mm
Coil material	copper	-

**Table 2 sensors-22-06065-t002:** Peak value of magnetic flux intensity under different lift-off values.

Magnetic Flux Intensity (nT)	Lift-Off Value of H (cm)				
	H = 0	H = 5	H = 10	H = 15	H = 20
Tangential component	1.81 × 10^−4^	4.65 × $10^{-5}$	1.85 × $10^{-5}$	9.44 × $10^{-6}$	6.25 × $10^{-6}$
Normal component	3.70 × 10^−4^	1.07 × 10^−4^	4.09 × 10^−5^	1.96 × 10^−5^	1.01 × 10^−5^

**Table 3 sensors-22-06065-t003:** Peak values of magnetic flux intensity under different lift-offs.

Magnetic Flux Intensity (nT)	Lift-Off Value of H (cm)				
	H = 0	H = 5	H = 10	H = 15	H = 20
Tangential component	8.34 × 10^−7^	5.44 × 10^−7^	4.79 × 10^−7^	4.28 × 10^−7^	4.33 × 10^−7^
Normal component	9.92 × 10^−6^	6.76 × 10^−7^	4.76 × 10^−7^	2.98 × 10^−7^	1.88 × 10^−7^

## Data Availability

Not applicable.

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
