# Peer review of "Propagation Characteristics of Magnetic Tomography Method Detection Signals of Oil and Gas Pipelines Based on Boundary Conditions"

_sensors, 2022, doi:10.3390/s22166065_

Round 1
Reviewer 1 Report
In this paper, numerical simulation and experimental study are adopted to investigate the variation law of the magnetic signal outside the pipe consideration two boundary conditions. Further improvement as listed below:
1. please provide more details about Villarreal effect.
2. please explain the physics terms, such as uxx, uyy, uzz in Eq.(6), and a in Eq.(16)-(20)
3. there are some typo errors should be corrected, such as in section 2 in line 198; the label of Fig.4; is the detection heights in 234 is the liftoff value, if yes, please correct.
4. it is suggested to list the geometric and electromagnetic parameters in a table. More detail of the simulation should provide to help reader to understand this paper. Such as what is the material of the pipe and its conductivity? what is the input magnetic field? please illustrate the boundary conditions for Fig.5(a) and Fig.5(b) respectively in FEM simulation? how the stress is simulated in the FEM model; please provide the information about the coil.
5. as that shown in Fig.6(a), the peak value is decreasing with the increasing lift off, why the Tangential component of Magnetic flux intensity is same for the different lift-off?
6. in Fig.7 and Fig.9, what is the Y axis, normal component of magnetic flux density or magnetic flux density? it is conflicting, why?
7. compared with Fig.6, the curves in Fig.8 is more complicated, why?
8. the inner diameter of pipe in experiment and simulation is quite different, could you keep them same?
9. you are suggested to provide a hardware of the experiment setup and provide more details.
10. compared with Fig.6 and Fig.11, the number of Y axis is an order of magnitude difference, what is the reason?
11. there are three data in Fig.13(d) and Fig.14(d), which is not enough to shown a nonlinear relationship. therefore, more experiment data is required.
Reviewer 2 Report
In my opinion, the adoption of a current-fed coil in the computational and experimental model as a stress concentration zone is incorrect, and therefore the information contained in the article adds nothing.
Reviewer 3 Report
The paper is well written. I suggest some improvement.
Page 2: I. Introduction: line 48: The measurement of very weak magnetic fields and the compensation of very small interfering magnetic influences is also very well described in methods that use a highly sensitive quartz sensor as the sensitive element. These methods have very high sensitivity and simultaneous conversion of the magnetic field into a frequency signal, as described in ref.:
- - Detection principles of temperature compensated oscillators with reactance influence on piezoelectric resonator. Sensors. 2020, vol. 20, iss. 3, p. 1-18. ISSN 1424-8220. https://www.mdpi.com/1424-8220/20/3/802
- Reduced Graphene Oxide with Special Magnetoresistance for Wireless Magnetic Field Sensor. Nano-Micro Letters 2020, 12, 1-14, doi:10.1007/s40820-020-0403-9.
- High resolution switching mode inductance-to-frequency converter with temperature compensation. Sensors, ISSN 1424-8220, 2014, vol. 14, no. 10, p. 19242-19259. https://www.mdpi.com/1424-8220/14/10/19242
Page 10: 4. 1 Design of experiment: line 311: Figure 10: Which detection sensor is used? What is the output of the sensors: voltage, currrent, frequuency or ????
Page 12: line 336. Figure 10: Where are positions 1 and 2?
Round 2
Reviewer 1 Report
Authors has already answered all the concerns of the reviewer.
Reviewer 2 Report
no comments